# Using Functional Connectivity to Examine the Correlation between Mirror Neuron Network and Autistic Traits in a Typically Developing Sample: A fNIRS Study

**DOI:** 10.3390/brainsci11030397

**Published:** 2021-03-20

**Authors:** Thien Nguyen, Helga O. Miguel, Emma E. Condy, Soongho Park, Amir Gandjbakhche

**Affiliations:** Eunice Kennedy Shriver National Institute of Child Health and Human Development, National Institutes of Health, 49 Convent Drive, Bethesda, MD 20892-4480, USA; thien.nguyen4@nih.gov (T.N.); helga.miguel@nih.gov (H.O.M.); emma.condy@nih.gov (E.E.C.); soongho.park@nih.gov (S.P.)

**Keywords:** action-observation, action-execution, superior parietal, inferior parietal, supramarginal, angular

## Abstract

Mirror neuron network (MNN) is associated with one’s ability to recognize and interpret others’ actions and emotions and has a crucial role in cognition, perception, and social interaction. MNN connectivity and its relation to social attributes, such as autistic traits have not been thoroughly examined. This study aimed to investigate functional connectivity in the MNN and assess relationship between MNN connectivity and subclinical autistic traits in neurotypical adults. Hemodynamic responses, including oxy- and deoxy-hemoglobin were measured in the central and parietal cortex of 30 healthy participants using a 24-channel functional Near-Infrared spectroscopy (fNIRS) system during a live action-observation and action-execution task. Functional connectivity was derived from oxy-hemoglobin data. Connections with significantly greater connectivity in both tasks were assigned to MNN connectivity. Correlation between connectivity and autistic traits were performed using Pearson correlation. Connections within the right precentral, right supramarginal, left inferior parietal, left postcentral, and between left supramarginal-left angular regions were identified as MNN connections. In addition, individuals with higher subclinical autistic traits present higher connectivity in both action-execution and action-observation conditions. Positive correlation between MNN connectivity and subclinical autistic traits can be used in future studies to investigate MNN in a developing population with autism spectrum disorder.

## 1. Introduction

The mirror neuron network (MNN) consists of a group of brain regions that are activated when individuals perform an action and observe others performing the same action [1]. Mirror neurons were originally discovered in non-human primates via single cell recording of neurons in area F5 of the ventral premotor cortex and inferior parietal lobe during both the performance and observation of a specific action [2,3]. Since then, studies have examined the same processes in the human brain and attempted to establish links between action execution, recognition and perception, and higher cognitive processes. Findings from human neuroimaging studies using functional magnetic resonance imaging (fMRI) and functional near-infrared spectroscopy (fNIRS) show a similar key set of regions that are active during action observation and action execution namely inferior frontal gyrus, ventral premotor cortex and inferior parietal lobe [4,5,6,7,8,9,10]. However, other brain regions have also been involved, namely the supplementary motor, posterior middle temporal gyrus or primary visual cortex [5].

Traditionally, analysis of the MNN is conducted by analyzing activation across regions-of-interest (ROI) to characterize which brain regions overlap during observation and execution of an action, but this approach has limitations [11]. ROI analysis focuses on the magnitude of activation but does not provide information about the temporal dynamics of brain activation, or the way spatially distant regions are connected during cognitive processes. It is likely that action-execution and action-observation processes comprise a complex interplay between different regions in the brain. To address the interaction between brain regions, using an analysis approach like functional connectivity in the MNN may be more informative as it provides information about the way brain regions co-activate during a task. Previous MNN studies using fMRI have shown that parietal and temporal connectivity decrease when observing familiar actions [12] and increase with task complexity [13]. In addition, functional connectivity in the rostral inferior parietal lobe was found to be positively correlated with age [14], suggesting that the MNN develops and is reorganized with experience and maturation. These conclusions support that functional connectivity might be an important contribution for the understanding of the functional relationships between MNN nodes.

It has long been debated whether dysfunction of the MNN underlies differences in social abilities. Representing and understanding the actions of others as our own actions has been linked to the development of sophisticated social behaviors such as imitation, theory of mind (ToM), or empathy [15]. This is particularly relevant for the study of populations that lack social reciprocity, for example individuals with autism spectrum disorder (ASD). Studies have shown that the MNN network in individuals with ASD seem to be different from healthy individuals, particularly during action-observation conditions [16,17]. Most studies have used traditional fMRI ROI analysis to identify brain regions that are associated with action-observation in the ASD population [18]. Others have examined it using functional connectivity and reported a mixed result in this population. While Cole et al. found a reduction in connectivity between mentalizing networks and the MNN [19], Fishman et al. reported an increase in connectivity between the regions of the MNN and ToM [20] in ASD population. In addition, Martino et al. found that the relationship between functional connectivity and autistic traits in neurotypical adults was negative in anterior mid-insula, but positive in angular gyrus and superior parietal cortex [21]. One approach that has yielded additional information about brain regions that may relate to deficits seen in psychopathologies, such as ASD, has been the identification of areas associated with subclinical traits in typical populations [22]. Indeed, in taking a dimensional approach to the study of cognitive processes and disorders, the study of subclinical traits or the “full range of variation, from normal to abnormal” [23] is being encouraged in psychiatric research through initiatives such as the Research Domain Criteria (RDoC). For example, subclinical autistic traits in typically developing children are shown to relate to structural neuroimaging metrics, such as decreased cortical gyrification in the left temporal area [24] and cortical thinning in right superior temporal sulcus [25]. The aim of the present study is to expand this approach by examining connectivity of the MNN in relation to autistic traits in a normative adult population to determine if similar trends can be detected.

In the current study, we propose the use of fNIRS for the study of MNN in the context of a live paradigm in which the individual produces an everyday action (reaching, grasping, and moving a cup and observing someone performing the same action). The purpose of this study is to assess MNN connectivity in adults and establish a method that can be used in future research investigating the MNN in populations at-risk for ASD. The aim of this study is threefold: (1) to demonstrate that fNIRS can be applied to the study of the MNN; (2) to identify cerebral functional connections that are specific for the MNN; (3) to conduct a preliminary examination of the relation between MNN connectivity and autistic traits in a sample of healthy volunteers. We hypothesize that the MNN paradigm will result in a strong functional connectivity within the inferior parietal lobe and between brain regions in the parietal lobe during observation and execution of an action as measured through fNIRS. In addition, we hypothesize that individuals with more autistic traits will present greater connectivity in these brain regions.

## 2. Materials and Methods

### 2.1. Participants and Experimental Protocol

The experiment was performed at the National Institutes of Health (NIH) and all procedures were conducted in accordance with the guidelines and regulations of the Institutional Review Board in NIH (protocol number: 18-CH-0001). A total of 35 healthy adults (19 female, mean age = 33.8) volunteered to participate in the study. An additional 5 participants were tested but excluded due to noisy data (*n* = 2), experimental error (*n* = 2), or not having the minimum number of good trials (*n* = 1).

After signing the consent form, participants underwent a health assessment and completed the Autism-Spectrum Quotient (AQ) questionnaire. The AQ is a 50-item self-report questionnaire designed to assess the degree to which an adult demonstrates behavioral and cognitive traits associated with the autism spectrum [26]. The AQ has been used in normative samples [27] as it provides a dimensional measure of ASD traits across both the clinical and normative range. Furthermore, a measurement invariance analysis indicated that the short form of the ASQ, which takes 28 items from the full AQ and is highly correlated with the full version [28] measured the same construct across an ASD sample and typical sample [29] though scalar invariance was not attained, which may make comparison of AQ scores between a normative and clinical sample invalid. However, as a continuous measure, the AQ has been shown to be valid in nonclinical samples [30] and has been used as a correlate of biological substrates, such as heritability [31] and neural measures [32,33] in such populations.

The experiment consisted of two conditions: action-execution and action-observation. During the action-execution, a cup was presented in front of the participant for him/her to grasp, lift and move it toward himself/herself; during the action-observation, the participant observed an experimenter performing the same action. There were 15 trials in each condition. Each trial lasted approximately 30 s, including 5 s of task, 20 s of rest (when the participant observed a moving pendulum), and 5 s of task-rest transition when a vertical curtain was lowered to block the participant’s view as the cup was replaced with a pendulum. Trials from the conditions were randomized to prevent participant from predicting the next condition.

### 2.2. Data Recording

A 24-channel NIRS system (Hitachi ETG-4100) was used to record the cerebral hemodynamic response in the central and parietal cortex in both sides of the brain. The NIRS system consists of 18 optodes: eight sources emitting light at 695 nm and 830 nm and 10 detectors. The optodes were fixed with a 3D printed frame, which were attached to a 128-electrode electroencephalogram (EEG cap, Electrical Geodesic, Inc., Eugene, OR, USA) (Figure 1). EEG caps of different sizes were used based on each participant’s head circumference; as a result, there was minor variance in the inter-optode spacing (M = 2.88 ± 0.13 cm). The system sampling rate was 10 Hz. The optode location was recorded after the experiment using a 3D digitizer (Fastrak, Polhemus).

Data from the digitizer was used to assign a brain region to each NIRS channel using Atlas Viewer [34]. Due to the head size and shape difference, the brain regions were not consistent for all participants. A total of 28 brain regions ranging from superior frontal lobe to middle occipital area were assigned to NIRS channels with the number of brain regions varying by participant (Table 1). Analysis was performed on the regions that were present in more than 15 participants (50% of sample), resulting in twelve regions of interest (ROI) (6 regions in each hemisphere) were selected (Table 1, bold regions).

### 2.3. Data Analysis

#### 2.3.1. Hemodynamic Response and Functional Connectivity

Principle component analysis (PCA) was applied to measured signals to remove systemic and movement artifact. Figure 1b represents a power spectrum density of HbO data from channel 7 in subject 19 before and after PCA to demonstrate the effect of PCA in removing systemic artifact. The task time for the action-observation and action-execution condition was 5 s, however, the experimenter/participant completed the action in less time. Hence, the NIRS data were detrended and lowpass filtered at 0.5 Hz (not 0.1 Hz as in other resting state functional connectivity studies) to retain a possible fast brain response. Cerebral hemodynamic responses, including oxy- (HbO) and deoxy- (Hbb) hemoglobin, were derived from measured NIRS signal using the modified Beer-Lambert law with a differential pathlength factor of 6 for both wavelengths. Hemodynamic data were averaged across the 15 trials for each condition. Functional connectivity was then calculated for each channel pair as Pearson correlation coefficients (ρ) of HbO during each task. Data from HbO was used for connectivity calculations because it was shown to be more specific to the cerebral connectivity when performing a task [35]. The sampling distribution ρ values were transformed to the normal distribution z values using Fisher z-transformation [35,36]. Subsequent data analysis was conducted on these z values. Finally, region-to-region connectivity values for each participant were derived from channel-to-channel connectivity values by averaging all the connections belonging to that region-to-region pairing, as there were often multiple instances of the same region-to-region pairing within a participant. Further analyses were done entirely on region-to-region connectivity. A total of 78 region-to-region connections, with 12 intra-ROI connections and 66 inter-ROIs connections were computed.

#### 2.3.2. Statistical Analysis

Normal distribution z values were used for all statistical analyses. For each participant we averaged the connectivity of all region-to-region connections (78 connections) by condition, resulting in one average connectivity value per participant. The average connectivity was then compared to each region-to-region connectivity in each condition across all participants using a simple, two-tailed, paired t-test with a critical *p*-value of 0.05. The purpose of the test is to identify connections which have a connectivity value greater than the average connectivity within a condition. Due to differences in the number of participants having each ROI, separate statistical tests were conducted to analyze each region-to-region connection with the use of different degree of freedoms in each statistical test. A connection was considered to have greater connectivity in a condition when its connectivity was significantly higher than the average connectivity. The region-to-region connectivity of all region pairs in each condition was correlated with AQ score. The correlation was considered significant when the *p*-value was less than or equal to 0.05.

## 3. Results

### 3.1. Connectivity during Action-Execution and Action-Observation

The region-to-region functional connectivity across all participants during the action-execution and action-observation is displayed in Figure 2a,b. Comparison of each region-to-region connectivity value with the average connectivity in each condition reveals seven statistically significant connections (five intra-ROI and two inter-ROI) in the action-execution condition (red nodes, edges, Figure 2a) and 10 statistically significant connections (seven intra-ROI and three inter-ROI) in the action-observation condition (red nodes, edges, Figure 2b). Among connections with significantly greater connectivity, there are five significant connections (four intra-ROI and one inter-ROI) that overlapped between the two conditions: connections within the right precentral, right supramarginal, left inferior parietal, left postcentral, and between the left supramarginal and left angular regions (Figure 2a,b).

### 3.2. Correlation between Connectivity and Autistic Traits

Subclinical autistic traits measured in 30 participants resulted in AQ score ranging from 4 to 30 with an average of 14.07 (±7.00). Pearson correlations were conducted between AQ score and region-to-region connectivity in each condition. Figure 3 and Table 2 display the relation between the AQ score and the connectivity in connections with significant correlation coefficients. In general, there exists a positive, linear correlation between connectivity values and AQ score, which means that an individual with higher autistic traits has higher connectivity between these regions (Figure 3a,b).

## 4. Discussion

Traditional analysis of the human MNN has mainly focused on identifying ROIs related to executing an action or observing someone performing an action. To the best of our knowledge there are no existing studies on MNN connectivity during a live action-observation and action-execution task. In our study we found that connections in the right precentral, right supramarginal, left inferior parietal, left postcentral, and between left supramarginal-left angular regions seem to be highly involved in the MNN. Additionally, our preliminary analysis on MNN connectivity and autistic traits suggests that typically developing individuals with greater autistic traits as measured by the AQ present greater connectivity within the MNN during both execution and observation of actions.

In the current study we examined functional connectivity for action execution, action observation and explored MNN connectivity by identifying connections which have significantly greater connectivity in both conditions. Our results showed that during the action execution, while a participant was performing an action with the right hand several region-to-region connections within the left hemisphere were related (connections within the left precentral, left postcentral, left inferior parietal and between the left supramarginal and left angular regions), which is consistent with the contralateral nature of pyramidal tracts [37]. Interestingly, the observation condition resulted in a greater number of bilateral connections than the execution condition (action observation: five inter-left and five inter-right hemisphere connections, action execution: four inter-left and three inter-right hemisphere connections). Traditional MNN analysis using fNIRS has shown that action observation elicits more bilateral activation when compared to execution [38], therefore it is not surprising that this is also captured through connectivity analyses. Additionally, the connections that have significantly greater connectivity in both conditions were connections within the left inferior parietal, right supramarginal and between the left supramarginal and left angular regions, which is consistent with previous findings of the link between bilateral inferior parietal lobe and the MNN [20]. A fundamental difference between our study and previous studies is that we computed functional connectivity for a task-based approach, where other studies have computed connectivity during a resting state. In this regard we believe our approach adds to the current literature on the MNN as it allows us to better understand how brain regions interact and co-activate during the task. In general, our study’s results have proved the feasibility of using functional connectivity derived from fNIRS signal as a new avenue to investigate the MNN.

Our preliminary analysis on the relation between autistic traits as measured by the AQ and MNN showed a positive linear correlation between MNN connectivity and subclinical autistic behaviors in a sample of healthy individuals. In accordance with previous studies [20,21] and in line with what we hypothesized, participants with more autistic traits presented greater functional connectivity when performing an action and observing someone performing an action. Specifically, parietal regions, namely inferior and superior parietal, angular and supramarginal cortex, resulted in stronger correlations with autistic traits. Interestingly, these regions have been identified as having abnormal functional connections with higher order brain regions, such as the medial and dorsolateral prefrontal cortex [19,20] in a population with ASD. These abnormalities might underlie the difficulties this population has in understanding the internal states from their own actions and actions of others. Furthermore, traditional analyses of the MNN by ROI have shown a greater activation in the inferior parietal cortices (angular and supramarginal gyri) in ASD children compared to typically developing children during action observation, execution, and interpersonal synchrony tasks [39]. In general, the elevation of both functional connectivity and brain activation in the MNN of the individuals with ASD might reflect a mechanism to compensate for social deficits [20]. Another finding from this study is that a greater number of connections, which have significant correlation between connectivity and AQ, was observed in the action-observation (13 connections) compared with the action-execution (five connections). Though this result could be related to the fact that observing others performing an action (10 connections) recruited more connections than performing the action (seven connections), further studies need to be conducted to verify the finding. Generally, a positive correlation between MNN connectivity and autistic traits suggests a potential application of the NIRS based connectivity approach in studying developmental populations, namely children at-risk for ASD.

In this study, fNIRS, which has several advantages compared to other neuroimaging techniques especially when studying motor tasks [38], was used to examine the MNN. Compared to fMRI, fNIRS has better temporal resolution, is more robust to movement artifact, and is ecologically valid, making it easy to use with developing and clinical population. The current study is part of a broader study to examine the MNN in a developing population, with whom fMRI is not feasible. In addition, the goal of this study was to investigate functional connectivity during an action-execution task, which would be difficult to conduct with fMRI due to the high level of movement artifact associated with the action-execution condition.

Findings from this study extend our knowledge on the neural activity of the MNN and they suggest a new approach to study the MNN in general and the MNN in high-risk ASD children in specific. However, our study is not without limitations. Current NIRS probes only cover the central and parietal cortex, hence we were not able to study other brain regions of the MNN such as ventral premotor cortex in the prefrontal lobe or posterior middle temporal gyrus in the temporal lobe or primary visual cortex in the occipital lobe. For future studies, we suggest using the whole head fNIRS system so that the MNN connectivity can be examined completely. Additionally, autistic traits and social communication skills were only evaluated through a single self-report measure. While this measure has been widely used in typically developing samples, the reliance on one measure is a notable limitation, as deeper behavioral phenotyping could provide a more specific idea from where the relationships between these traits and functional connectivity are derived. Future studies attempting to evaluate correlates of this behavioral construct in typically developing samples should aim to thoroughly assess their participants in an atypical population.

## 5. Conclusions

This study investigated the cerebral functional connectivity in the central and parietal cortex during a live action-observation and action-execution to explore the mirror neuron network (MNN) connectivity. Connections within the right precentral, right supramarginal, left inferior parietal, left postcentral, and between the left supramarginal-left angular regions were found to be candidate connections of the MNN connectivity. In addition, region-to-region connectivity in action-observation and action-execution was correlated to the autism-spectrum quotient score to examine relationship between connectivity and autistic traits. An individual with higher autistic traits presents a greater connectivity in both conditions. Findings from this study can be used to assess autistic traits in the children at-risk for autism spectrum disorder.

## Figures and Tables

**Figure 1 brainsci-11-00397-f001:**
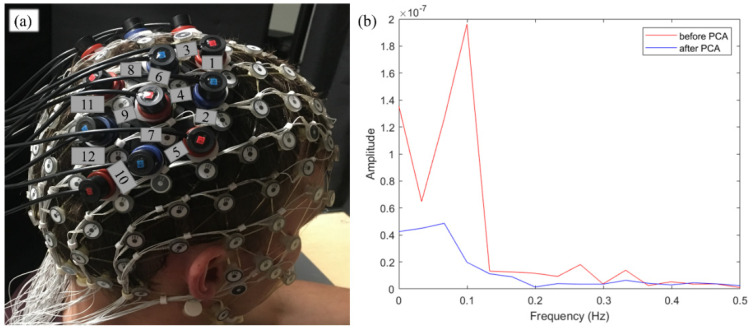
(**a**) NIRS optode arrangement. Red optodes: light sources; blue optodes: light detectors; numbers: channels formed by a source-detector pair; (**b**) power spectrum density analysis on hemodynamic signals before and after principle component analysis (PCA).

**Figure 2 brainsci-11-00397-f002:**
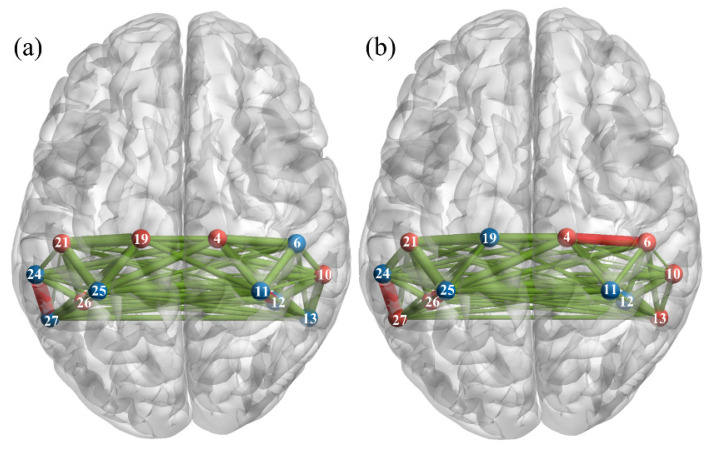
Mean connectivity during (**a**) Action-Execution, (**b**) Action-Observation. The edge thickness represents the connectivity strength. Red edge, node: connections which have significantly greater connectivity.

**Figure 3 brainsci-11-00397-f003:**
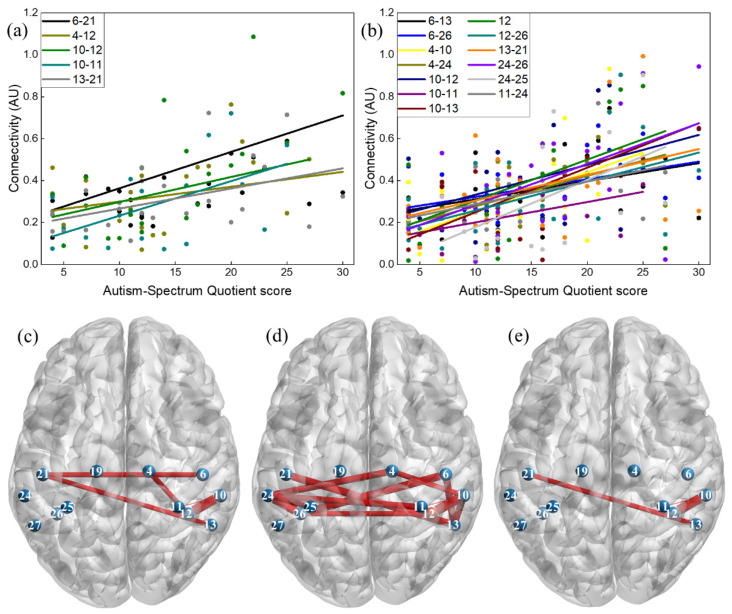
Correlations between functional connectivity and AQ score: (**a**) 5 connections with a significant correlation coefficient during Action-execution; (**b**) 13 connections with a significant correlation coefficient during Action-observation; and brain maps showing the location of connections with significant correlation coefficients (**c**) during action-execution; (**d**) during action-observation; (**e**) during both conditions.

**Table 1 brainsci-11-00397-t001:** Brain regions measured with NIRS. Bold regions: selected regions of interest (ROI); Hem: hemisphere, Par: Number of participants. The numbers in the ROI column are indices of the ROIs.

Brain Region	Hem	ROI	Par	Brain Region	Hem	ROI	Par
Superior frontal	Right	1	8	Superior temporal	Right	8	2
	Left	16	8	Middle temporal	Right	9	2
Middle frontal	Right	2	1		Left	23	1
	Left	17	4	**Supramarginal**	**Right**	**10**	**26**
Supplementary motor	Right	3	3		**Left**	**24**	**26**
	Left	18	1	**Superior parietal**	**Right**	**11**	**22**
**Precentral**	**Right**	**4**	**26**		**Left**	**25**	**19**
	**Left**	**19**	**27**	**Inferior parietal**	**Right**	**12**	**30**
Paracentral	Right	5	3		**Left**	**26**	**30**
	Left	20	13	**Angular**	**Right**	**13**	**30**
**Postcentral**	**Right**	**6**	**28**		**Left**	**27**	**27**
	**Left**	**21**	**29**	Superior occipital	Right	14	1
Precuneus	Right	7	1	Middle occipital	Right	15	6
	Left	22	4		Left	28	3

**Table 2 brainsci-11-00397-t002:** Connections which have a significant correlation between connectivity in a condition and AQ score. Regions in bold: connections overlapped between the two conditions.

**Execution**							
**Connection**	**r**	***p*-Value**	**Par**	**Connection**	**r**	***p*-Value**	**Par**
Right precentral— right inferior parietal	0.46	0.03	22	Right postcentral— left postcentral	0.46	0.02	25
**Right supramarginal—** **right inferior parietal**	**0.42**	**0.04**	**23**	**Right angular—** **left postcentral**	**0.44**	**0.02**	**26**
**Right supramarginal—** **right superior parietal**	**0.53**	**0.03**	**17**				
**Observation**							
Right postcentral— right angular	0.40	0.05	25	**Right angular—** **left postcentral**	**0.42**	**0.03**	**26**
Right supramarginal— right precentral	0.53	0.02	19	Right inferior parietal— left inferior parietal	0.51	0.01	27
**Right supramarginal—** **right inferior parietal**	**0.46**	**0.03**	**23**	Left supramarginal— right superior parietal	0.49	0.04	18
**Right supramarginal—** **right superior parietal**	**0.49**	**0.05**	**17**	Left supramarginal— right precentral	0.50	0.02	20
Right supramarginal— right angular	0.54	0.01	23	Left supramarginal— left inferior parietal	0.55	0.003	26
Right parietal inferior	0.54	0.03	17	Left supramarginal— left superior parietal	0.51	0.04	16
Right postcentral— left parietal inferior	0.45	0.02	26				

## Data Availability

The data presented in this study are available on request from the corresponding author. The data are not publicly available due to privacy restriction.

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
