# Peer review of "Using Functional Connectivity to Examine the Correlation between Mirror Neuron Network and Autistic Traits in a Typically Developing Sample: A fNIRS Study"

_brainsci, 2021, doi:10.3390/brainsci11030397_

Round 1

Reviewer 1 Report

Comments and Suggestions for Authors

This was a well-written manuscript that presented somewhat novel results regarding functional connectivity of brain regions associated with mirror neuron function and behavioral correlations between these regions and autistic traits in healthy adults. While the methods and analysis seem sound in general, some revisions are needed, especially concerning the background, hypotheses and interpretation of the results. Suggestions about such revisions are outlined below.

MAJOR REVISIONS:

  • The introduction had some possible areas of improvement, including the hypotheses. For instance, the connection between social functioning and the MNN was lacking a bit. The authors could make a better case for the association between mirror neuron function and its role in social development (e.g., imitation), for instance. More serious than this issue, however, were some weaknesses in the hypotheses. That is, the authors mentioned that they projected to observe “strong functional connectivity between inferior parietal lobe and surrounding brain regions” (italics added). The credibility of the hypothesis could be strengthened by the authors being more specific about which brain regions might be included in the above. Additionally, the authors hypothesize greater functional connectivity in people with a higher degree of autistic traits. In contrast, they talk about reductions of functional connectivity earlier in the introduction. Citing a study/studies that would lead them to project increased functional connectivity would improve the consistency between the background and hypothesis, strengthening the drive of the paper. This issue with consistency persists in the Discussion section.
  • In section 2.3.1, the authors indicate that they have computed 78 region-to-region connections, etc. However, they do not mention applying a multiple comparisons correction to the results of these analyses. I suggest that the reviewers use and report such a correction. Doing so will lend credibility to and may clarify the results of the study. A similar correction could be employed in the analysis described in section 2.3.2.
  • In Figure 3, it might be clearer to the reader if the authors included both scatter and brain plots for both conditions, as well as the overlap between the conditions. That said, some modification to the brain plots might be warranted, since the connectivity of some of the brain regions mentioned in the text (esp. precentral gyrus) is not readily apparent.
  • The authors could improve their efforts to interpret their findings of greater functional connectivity between MNN regions and its association with autistic traits. Based on what is written in the Discussion, it’s not clear what this finding might say about MNN function in autistic individuals or why it might be related to increased expression of autistic traits. This notion is especially true when one considers that the task described in the study involved reaching for a cup and observing the same—the authors would do well to comment on the possible connections between such a task and more complex actions/observations, such as social functioning.
  • Given the popularity, availability, and excellent spatial resolution of other neuroimaging methods, such as fMRI, the authors need to improve their justification for the use of fNIRS in the Discussion (esp. on lines 258-260).

MINOR REVISIONS:

  • 3, line 117: the word “vertically” should read “vertical”
  • 7, line 269: the authors write “… a more specific idea of where the relationships between these traits and functional connectivity are derived from.” This sentence should be re-worded to say, “… a more specific idea from where the relationships between these traits and functional connectivity are derived.”
  • References are numbered in the text, but not in the references list.

Author Response

This was a well-written manuscript that presented somewhat novel results regarding functional connectivity of brain regions associated with mirror neuron function and behavioral correlations between these regions and autistic traits in healthy adults. While the methods and analysis seem sound in general, some revisions are needed, especially concerning the background, hypotheses, and interpretation of the results. Suggestions about such revisions are outlined below.

 We would like to send a special thanks to the reviewer for your great comments and suggestions. We really appreciate it. Please find our answer below.

MAJOR REVISIONS:

1) The introduction had some possible areas of improvement, including the hypotheses. For instance, the connection between social functioning and the MNN was lacking a bit. The authors could make a better case for the association between mirror neuron function and its role in social development (e.g., imitation), for instance. More serious than this issue, however, were some weaknesses in the hypotheses. That is, the authors mentioned that they projected to observe “strong functional connectivity between inferior parietal lobe and surrounding brain regions” (italics added). The credibility of the hypothesis could be strengthened by the authors being more specific about which brain regions might be included in the above. Additionally, the authors hypothesize greater functional connectivity in people with a higher degree of autistic traits. In contrast, they talk about reductions of functional connectivity earlier in the introduction. Citing a study/studies that would lead them to project increased functional connectivity would improve the consistency between the background and hypothesis, strengthening the drive of the paper. This issue with consistency persists in the Discussion section.

Thank you for your great suggestions. Association between mirror neuron function and its role in social development such as imitation and theory of mind has been added to the introduction (lines 59 to 64, page 2). The hypothesis “strong functional connectivity between inferior parietal lobe and surrounding brain regions” has been changed to “strong functional connectivity within the inferior parietal lobe and between brain regions in the parietal lobe” (lines 94-95 page 2). In addition, a reference (ref 21), which reported a positive relationship between autistic traits and functional connectivity in the angular gyrus and parietal lobe has been added and the statement in the introduction has been modified accordingly (lines 66 – 72, page 2).

2) In section 2.3.1, the authors indicate that they have computed 78 region-to-region connections, etc. However, they do not mention applying a multiple comparison correction to the results of these analyses. I suggest that the reviewers use and report such a correction. Doing so will lend credibility to and may clarify the results of the study. A similar correction could be employed in the analysis described in section 2.3.2.

We understand the reviewer’s concern regarding multiple comparison corrections. Our approach was to find connections that have a connectivity value stronger than the average connectivity within a condition. Our intention was not to compare the connectivity values between the two conditions which would require multiple comparison corrections. We agree that this might be seen as a limitation hence a sentence has been added to section 2.3.2 to provide more details (line 181-182, page 5). 

3) In Figure 3, it might be clearer to the reader if the authors included both scatter and brain plots for both conditions, as well as the overlap between the conditions. That said, some modification to the brain plots might be warranted since the connectivity of some of the brain regions mentioned in the text (esp. precentral gyrus) is not readily apparent.

We agree that adding brain plots for each condition and the overlap between the conditions have made the significant connection more apparent. In the revised version, we separated Figure 3c into Figure 3c (action-execution), Figure 3d (action-observation), and Figure 3e (overlap between both conditions).

4) The authors could improve their efforts to interpret their findings of greater functional connectivity between MNN regions and its association with autistic traits. Based on what is written in the Discussion, it’s not clear what this finding might say about MNN function in autistic individuals or why it might be related to increased expression of autistic traits. This notion is especially true when one considers that the task described in the study involved reaching for a cup and observing the same—the authors would do well to comment on the possible connections between such a task and more complex actions/observations, such as social functioning.

The discussion has been modified to interpret our findings (line 263-265 and line 268 – 270, page 8).

5) Given the popularity, availability, and excellent spatial resolution of other neuroimaging methods, such as fMRI, the authors need to improve their justification for the use of fNIRS in the Discussion (esp. on lines 258-260).

We acknowledge that compared to fNIRS, fMRI has an excellent spatial resolution, and it can measure hemodynamic signals in deep brain regions. However, the advantages of fNIRS over other modalities, especially fMRI includes better temporal resolution, is more robust to movement artifact, and is ecologically valid, making it easy to use with developing and clinical population. This information has been added to the discussion (line 279-287, page 8)

MINOR REVISIONS:

  • 3, line 117: the word “vertically” should read “vertical”

The word has been changed.

  • 7, line 269: the authors write “… a more specific idea of where the relationships between these traits and functional connectivity are derived from.” This sentence should be re-worded to say, “… a more specific idea from where the relationships between these traits and functional connectivity are derived.”

The sentence has been changed.

  • References are numbered in the text, but not in the references list.

References were numbered in the submitted manuscript, but it was removed by the journal during the formatting process.

Reviewer 2 Report

Comments and Suggestions for Authors

This article used functional near-infrared spectroscopy (fNIRS) to examine the autistic traits relevant to functional connectivity within the mirror neuron network. The study was well designed and the manuscript was well written. However, several issues ought to be addressed before publication. 

1) My major concern is that the low pass filter (0.5 Hz) applied to the fNIRS data in the current study. As the authors also noted, other RSFC studies use bandpass filters with cutoff frequencies at 0.01-0.1 Hz. Even though the authors stated that they have used PCA to remove the motion artifacts and physiological noises, there could still noises left in the data, especially Mayer's wave (~0.1 Hz). Therefore, the authors should at least provide a PSD analysis on signals before and after the PCA filtering to show that the noises in the data are removed from the correlation calculation. 

2) Another issue is that the study used a different localization method, in which the different ROIs had a varied number of participants. This was probably fine in the individual-level analysis, however, the authors should provide more details of how they balanced the statistical power in the group-level analysis (statistics). 

2) In Table 1, what are the numbers under the ROI categories? Are they Broadmann areas or just the indices of the ROIs? Please specify. 

3) The authors should provide a reason for why only HbO data was analyzed. 

4) Line 156 – 158, the authors stated that they averaged the connectivity values, please specify which value was used for this average, is it the rho? Or the Fisher's transformed rho? 

5) In the statistical analysis section, please also specify which connectivity values were used for the t-test. The correlation coefficients need to be converted to Z-score to be t-tested. 

6) Please provide more details of how the statistical analysis was done. For example, the authors used paired t-test, but they have to specify whether it was applied to the action-execution/action-observation. 

Author Response

This article used functional near-infrared spectroscopy (fNIRS) to examine the autistic traits relevant to functional connectivity within the mirror neuron network. The study was well designed and the manuscript was well written. However, several issues ought to be addressed before publication. 

Thank you so much for your very supportive comments and suggestions. We revised our manuscript to address your comments/suggestions as below.

1) My major concern is that the low pass filter (0.5 Hz) applied to the fNIRS data in the current study. As the authors also noted, other RSFC studies use bandpass filters with cutoff frequencies at 0.01-0.1 Hz. Even though the authors stated that they have used PCA to remove the motion artifacts and physiological noises, there could still noises left in the data, especially Mayer's wave (~0.1 Hz). Therefore, the authors should at least provide a PSD analysis on signals before and after the PCA filtering to show that the noises in the data are removed from the correlation calculation. 

Thank you for pointing out a very important concern and your valuable suggestion. A plot showing the PSD analysis on hemodynamic signals before and after the PCA filtering has been added to Figure 1b. Before PCA, Mayer’s wave at around 0.1 Hz was dominant, but this wave was removed by PCA.   

2) Another issue is that the study used a different localization method, in which the different ROIs had a varied number of participants. This was probably fine in the individual-level analysis, however, the authors should provide more details of how they balanced the statistical power in the group-level analysis (statistics). 

The statistical power in the group-level analysis was balanced through the use of different degree of freedoms in each test. A sentence has been added to section 2.3.2 to provide more details (line 182-185, page 5).

3) In Table 1, what are the numbers under the ROI categories? Are they Broadmann areas or just the indices of the ROIs? Please specify.

The numbers under the ROI categories are just the indices of the ROIs. We now added this information to Table 1 legend (line 150, page 4).

4) The authors should provide a reason for why only HbO data was analyzed. 

A previous study found that while Hb is more specific to functional connectivity at rest, HbO is more specific to functional connectivity during a task (reference 33). In this study, we focus on functional connectivity when performing a task (observing and executing an action), hence HbO was used to analyze connectivity. We now added this clarification to section 2.3.1 (Lines 164-166, page 4).

5) Line 156 – 158, the authors stated that they averaged the connectivity values, please specify which value was used for this average, is it the rho? Or the Fisher's transformed rho? 

Fisher’s transformed rho was used for the average. A sentence specifying this information was added (line 168, page 4).

6) In the statistical analysis section, please also specify which connectivity values were used for the t-test. The correlation coefficients need to be converted to Z-score to be t-tested. 

Z-score (z values) was used for the t-test. This information is now specified at the beginning of the statistical analysis section (line 176, page 5).

7) Please provide more details of how the statistical analysis was done. For example, the authors used paired t-test, but they have to specify whether it was applied to the action-execution/action-observation. 

The paired t-test was applied separately to each condition. Average action-execution connectivity was only compared to action-execution connectivity of the other 78 region-to-region connections. The same thing was performed for the action-observation connectivity. A phrase “in each condition” and a sentence is now added to clarify this information (line 179, line 181-182, page 5).

Round 2

Reviewer 2 Report

Comments and Suggestions for Authors

The author fixed my comments and the quality of the manuscript improved.